Demographic rates of northern royal albatross at Taiaroa Head, New Zealand

Richard Yvan 1 yvan@dragonfly.co.nz
Perriman Lyndon 2
Lalas Chris 3
Abraham Edward R. 1
1 Dragonfly Science , Wellington , New Zealand
2 Department of Conservation , Moray Place, Dunedin , New Zealand
3 Department of Marine Science, University of Otago , Dunedin , New Zealand
Gandini Patricia
Electronic publication date: 2015 Apr 28
Publication date: 2015
Volume: 3
Electronic Location ID: e906
Received 2014 Dec 18; Accepted 2015 Apr 1
Copyright: © 2015 Richard et al.
Copyright year: 2015
Copyright holder: Richard et al.
License: This is an open access article distributed under the terms of the Creative Commons Attribution License, which permits unrestricted use, distribution, reproduction and adaptation in any medium and for any purpose provided that it is properly attributed. For attribution, the original author(s), title, publication source (PeerJ) and either DOI or URL of the article must be cited.
License URL: https://creativecommons.org/licenses/by/4.0/

Keywords: Northern royal albatross, Diomedea sanfordi, New Zealand, Age at first reproduction, Survival rate, Multi-state capture-recapture model, Population size, Recruitment, Population dynamics

Funding: New Zealand Department of Conservation’s Conservation Services Programme POP2011-09 This project was funded by the New Zealand Department of Conservation’s Conservation Services Programme (www.doc.govt.nz/csp) project POP2011-09, partially through a levy on the quota holders of relevant commercial fish stocks. The funders had no role in study design, data collection and analysis, decision to publish, or preparation of the manuscript.

==============================
Demographic rates, such as annual survival rate, are generally difficult to estimate for long-lived seabirds, because of the length of time required for this kind of study and the remoteness of colonies. However, a small colony of northern royal albatross (Diomedea sanfordi) established itself on the mainland of New Zealand at Taiaroa Head, making possible regular banding and monitoring of its individuals since the first chick fledged, in 1938. Data on the presence/absence of birds, as well as on breeding outcomes, were available for the period from 1989–90 to 2011–12, and included 2128 annual resightings of 355 banded individuals of known age. The main goal of the present study was to estimate the annual survival rate of juveniles, pre-breeders, and adults at Taiaroa Head. These rates were estimated simultaneously in a single Bayesian multi-state capture-recapture model. Several models were fitted to the data, with different levels of complexity. From the most parsimonious model, the overall annual adult survival rate was estimated as 0.950 (95% CI [0.941–0.959]). In this model, adult survival declined with age, from 0.976 (95% CI [0.963–0.988]) at 6 years, the minimum age at first breeding, to 0.915 (95% CI [0.879–0.946]) at 40 years. Mean annual survival of pre-breeders was 0.966 (95% CI [0.950–0.980]), and 0.933 (95% CI [0.908–0.966]) for juveniles. There was no discernible difference in survival between males and females, and there was no apparent trend in survival over time. Estimates of other demographic rates were also obtained during the estimation process. The mean age at first return of juveniles to the colony was estimated as 4.8 years (95% CI [4.6–5.1]), and the mean age at first breeding as 8.9 years (95% CI [8.5–9.3]). Because all the birds of the colony were banded, it was possible to estimate the total population size. The number of northern royal albatross present annually at the Taiaroa Head colony has doubled since 1989–90, and the current total population size was estimated to be over 200 individuals. The ratio of the total population size to the number of annual breeding pairs varied from 5 to 12 among years, with an overall mean of 7.65 (95% CI [7.56–7.78]), and this high variability highlights the need for a sufficient number of surveys of seabird breeding populations before reliable conclusions on population trends can be made. Although long-term data allowed estimates of demographic rates of northern royal albatross at Taiaroa Head, the location of the colony and the ongoing management by staff mean that the population dynamics may differ from those of the main population on the Chatham Islands.

Introduction

Northern royal albatross, Diomedea sanfordi (Murphy, 1917), is a species of great albatross that breeds only in New Zealand (Agreement on the Conservation of Albatrosses and Petrels, ACAP, 2009). The species is classified as “Endangered” by the International Union for Conservation of Nature (IUCN, 2013), on the basis of a restricted breeding range and a projected rapid decline due to low productivity caused by severe storms in the 1980s. The New Zealand Threat Classification System lists this albatross as “Naturally Uncommon” (Miskelly et al., 2008).

Like many albatross species (IUCN, 2012), northern royal albatross are captured in commercial fisheries, including those using trawl and longlining methods (Abraham, Thompson & Berkenbusch, 2013). Fisheries bycatch has been linked to population declines in a number of seabird species including albatross (Lewison & Crowder, 2003; Baker et al., 2007; Tuck et al., 2001). The total population size of northern royal albatross is 5200 annual breeding pairs (ACAP, 2009). While the majority (99.5%) of the population breeds on the Chatham Islands, a small colony has become established at Taiaroa Head (45.7748 °S, 170.7278 °E), at the tip of Otago Peninsula, on the mainland of South Island, New Zealand. All individuals of this population have been monitored since its inception, with the first chick fledging in 1938 (Fleming, 1984).

Demographic information from the main breeding colonies of northern royal albatross is sparse. Although there is some knowledge of the reproductive biology of this species, and available data include estimates of adult survival (ACAP, 2009), detailed data are lacking, as these colonies are remote and difficult to access on a regular basis. For this reason, information from the colony at Taiaroa Head provides a unique opportunity to investigate, in detail, the demographic parameters that influence population trajectories (e.g., adult survival rate and age at first reproduction; Caswell, 2001; Beissinger & McCullough, 2002).

Northern royal albatross breed biennially, i.e., adults breed every second year when successfully producing a fledgling (ACAP, 2009). The breeding season starts in October to November, and breeding pairs lay a single egg in October to December. Eggs hatch mostly between late January and early February, and chicks fledge between August and October. Juveniles do not return to the colony until they are at least three years old, and first breeding has been reported at around eight years of age on average (IUCN, 2012).

The colony area at Taiaroa Head has been administered as a Nature Reserve since 1964, first managed by the New Zealand Wildlife Service then by the Department of Conservation (Robertson, 1993). Since 1968, routine monitoring has included (almost) daily visits by staff throughout the year to record the activity and status of individuals, and to carry out predator control (Robertson, 1993). The latter involves an intensive trapping programme that targets introduced predators—stoats Mustela ereminea, ferrets Putorius putorius, and cats Felix catus—at the colony.

All birds in the Taiaroa Head colony have been banded systematically since 1938, and some individuals have now been monitored for several decades (Robertson, 1993). Data up to 1993 have been used previously to estimate survival rates and longevity of the species (Robertson, 1993), but these estimates may not reflect the current population status. For example, potential threats to northern royal albatross may have changed since the estimates were derived, including the distribution of fishing effort (Abraham, Thompson & Berkenbusch, 2013), and climate and marine conditions (Doney et al., 2012). Furthermore, modern quantitative methods, such as Bayesian and multi-state modelling provide improved techniques for the estimation of demographic parameters, providing the impetus for the present study.

This study used an integrative Bayesian multi-state capture-recapture model to estimate the population size, annual survival rates, age at first return to the colony, and age at first reproduction of northern royal albatross. The model was based on resighting data recorded at Taiaroa Head from 1989–1990 to 2011–2012.

Methods

Dataset

Data used in this study included resightings of banded individuals in the period from 1989–90 to 2011–12. Data consisted of the presence/absence of each individual at the colony at the beginning of the breeding season (November), with resightings pooled over two months to ensure that the status of birds was established with certainty. The outcome of each bird’s breeding activity for each breeding season was also recorded, using codes indicating whether the bird attempted to breed or not, and whether the breeding attempt was successful (chick fledged) or not. These resightings data were summarised for the number of pre-breeders, breeding adults, and non-breeding adults present at the colony each year, and also for the number of chicks produced (Fig. 1).

Figure 1 Number of chicks produced, number of pre-breeders, breeding adults, non-breeding adults, and all individuals of northern royal albatross present at the Taiaroa Head colony each year from 1989–90 to 2011–12.

The number of chicks produced and the total number of birds at the colony in 2011–12 were unknown as the season was not complete that year.

Also included in the data were anecdotal reports of birds recovered dead. Ten birds were found dead at or near the Taiaroa Head colony, and two fatal captures in pelagic longline fisheries off Uruguay were also reported (S Jiménez, pers. comm., 2012).

The age of 355 of 382 individuals in the available dataset was known because the majority of birds at the colony was banded at the colony in their first year as chicks. Although resighting data were not available prior to 1989–90, the year of fledging and the gender of individuals banded before this season were available. The 27 individuals of unknown age were immigrants or visitors from the Chatham Islands population, and were not included in the analysis. These individuals were most likely juveniles prospecting for potential breeding sites. Among them, 10 individuals were only seen once, whereas 17 birds became established and bred at Taiaroa Head between 1989–90 and 2011–12.

The gender of each individual was first determined from measurements made during banding, as the distributions of wing and bill lengths are generally non-overlapping between males and females; males are larger than females (de L Brooke, 2004a). The gender was subsequently confirmed by the breeding behaviour of birds observed as adults. The gender was unknown for 34 (9.6%) of the 355 birds included in the analysis, and was in these cases treated as missing value.

In addition to the intensive trapping of introduced predators, breeding success at the Taiaroa Head colony is manipulated by fostering out an egg or chick from an adult that has presumably lost its partner. Chicks are weighed regularly to determine weight loss, and the latter may lead to supplementary feeding to ensure the survival of these chicks through winter. Abandoned eggs are fostered to other nests or artificially incubated until a spare nest becomes available. Also, care is given to chicks suffering from fly infestation or heat stress (by spraying the colony with water). This intervention has led to an increase in fledging rate from 54% to 74%, and the colony has steadily increased in size since its inception (Robertson, 1993). The annual number of individuals at the colony doubled from 1989–90 to 2011–12, from 68 individuals recorded at the colony in 1989–90 to 128 individuals in 2010–11 (Fig. 1).

The final dataset included 2128 annual resightings of 355 banded individuals of known age.

Modelling

To estimate the annual local survival rate of northern royal albatross at the Taiaroa Head colony, a Bayesian multi-state capture-recapture model was developed, adapted from a model for Gibson’s albatross Diomedea antipodensis gibsoni (Dillingham et al., 2012). Multi-state capture-recapture models (Brownie et al., 1993; Nichols & Kendall, 1995; Mijeom & Pollock, 2002; Kendall & Nichols, 2002) allow simultaneous estimations of the survival rate of different life history stages, while recognising that individuals might be alive but not recorded in a given year. Birds may not be recorded as they may be at the colony without being seen, or they did not return to the colony that year, as is typical behaviour in adults that successfully produced a fledgling the previous year (Robertson, 1993). In addition, the same model can estimate the probabilities of moving between life history stages, from which estimates of the age at first return and age at first reproduction can be easily obtained.

A Bayesian framework using the BUGS language (Spiegelhalter et al., 2003) provides a natural, transparent and flexible environment for implementing such fundamentally complex models, and for data with unobservable states, such as successful breeders that are not at the colony in a particular year. Multi-state capture-recapture models may be fitted in a non-Bayesian way (e.g., Gimenez et al., 2012).

Three life history stages were recognised, and survival was estimated for each stage: juveniles (from fledging to the first return to the colony), pre-breeders (from first return to first breeding at the colony), and adults (after the first breeding). In a given breeding season (12 months from November, called hereafter “year”), the state of each individual was assigned one of five possible and mutually exclusive states. There were four live states—juvenile (J), pre-breeder (PB), breeding adult (B), non-breeding adult (NB)—and one dead state (D)(see Fig. 2 for a diagram of the base model representing the different states and the possible transitions between them).

The intensive monitoring and management of the entire colony, being visited at least daily (with few exceptions) since 1968, and the relatively small colony size, means that the detection of individuals present at the colony is almost perfect, and we assumed here that the detection probability is 1 for individuals present at the colony.

Figure 2 Diagram of the Bayesian multi-state capture-recapture model used to estimate the survival rate of northern royal albatross at Taiaroa Head.

Shown are the five possible states of individuals and the annual transitions between them (J, juvenile; PB, pre-breeder; B, breeding adult; NB, non-breeding adult; D, dead).

Transitions between states, i.e., the probability of changing from one state to another in a given time step, were modelled explicitly. The probabilities of changing to a particular state given the previous state were: (1) PPBt|Jt−1=RaϕJ

(2) PBt|PBt−1=BaϕPB

(3) PBt|NBt−1=Pbreed|non-breederϕA

(4) PNBt|Bt−1=1−Pbreed again|successϕAafter a successful breeding attempt1−Pbreed again|failϕAafter a failed breeding attempt,

where t is the breeding season, ϕ{J,PB,A} the annual survival rate of juveniles, pre-breeders, and adults, respectively, Ra the probability of a juvenile of age a returning to the colony, Ba the probability of a pre-breeder of age a breeding for the first time, P(breed|non-breeder) the probability of an adult breeding in a particular year given it was a non-breeding adult the previous year, P(breed again|success) the probability of an adult breeding in a given year when it was a successful breeder the previous year, and P(breed again|fail) the probability of an adult breeding in a given year when it was a failed breeder the previous year.

The probabilities of remaining in the same live state from one year to the next were: (5) PJt|Jt−1=1−RaϕJ

(6) PPBt|PBt−1=1−BaϕPB

(7) PNBt|NBt−1=1−Pbreed|non-breederϕA

(8) PBt|Bt−1=Pbreed again|successϕAafter a successful breeding attemptPbreed again|failϕAafter a failed breeding attempt.

The probabilities of being dead (D) in a particular year, given the previous state were: (9) PDt|Jt−1=1−ϕJ

(10) PDt|PBt−1=1−ϕPB

(11) PDt|Bt−1=1−ϕA

(12) PDt|NBt−1=1−ϕA

(13) PDt|Dt−1=1.

The probabilities of all other transitions between states were set to 0.

The probability Ra of a juvenile of age a returning to the colony and becoming a pre-breeder was set to 0 at ages below the minimum observed age at first return (3 years; Robertson, 1993). This probability was then estimated independently each year for birds aged from three to seven years old. From eight years of age, this probability was fixed at a constant value, to represent the long tail in the distribution of age at first return that was observed in the data (i.e., the finding that a small number of birds take a long time to return compared with the majority of individuals): (14) Ra=0for 1≤a<3Ra=Pfirst return|age=afor 3≤a<8Ra=Pfirst return|age≥8for a≥8.

Similarly, the probability Ba of a pre-breeder of age a breeding for the first time was defined as follows, as the minimum age at first reproduction observed at Taiaroa Head was six years and the observed distribution of the age at first reproduction also exhibited a long tail (i.e., some birds take a long time to breed or they never breed): (15) Ba=0for 1≤a<6Ba=Pfirst breeding|age=afor 6≤a<11Ba=Pfirst breeding|age≥11for a≥11.

Simultaneously to modelling the state of individuals, the probability of individuals being at the colony in a given year was also modelled, with each individual being either present at the colony at least at the beginning of the breeding season, or not (either spending the year elsewhere or dead). The probability of being at the colony P(Ct|Xt) was modelled conditionally on the state X (juvenile, pre-breeder, breeding adult, non-breeding adult, or dead) in a given year t. Breeding adults were at the colony by definition. Similarly, juveniles were categorically not at the colony. The probability of being at the colony was independently estimated between pre-breeders, non-breeders who bred successfully the previous year, and other non-breeders (e.g., non-breeders who had failed the previous year, or in an earlier year and had not bred since), with associated estimated parameters γPB, γNB|success, and γNB|fail, respectively: (16) PCt|Jt=0

(17) PCt|PBt=γPB

(18) PCt|NBt=γNB|success,for adults who bred successfully the previous year

(19) PCt|NBt=γNB|fail,for adults who did not breed successfully the previous year

(20) PCt|Bt=1

(21) PCt|Dt=0.

For the annual probabilities of first return and of first breeding, as well as the probabilities of being at the colony, a non-informative uniform prior between 0 and 1 was used. In the base model, a uniform prior between 0 and 1 was used for the survival rate of juveniles and pre-breeders, and between 0.7 and 1 for adults.

The probability of a breeding attempt failing or succeeding was not modelled, because breeding success in the Taiaroa Head colony is manipulated. Instead, the observed outcome of each breeding attempt as perceived by each individual bird was used as a covariate. For instance, if an egg was transferred from pair A to pair B, pair A was recorded as failed breeders while pair B was considered successful breeders (given the egg hatched and the chick fledged). Because the probability of success of a breeding attempt was not modelled, knowledge of this probability was required at each time step. Consequently, all breeding adults were assumed to be at the colony; adults not seen at the colony in a given year were assumed to be non-breeders in that year. This assumption was deemed reasonable, based on the very high fidelity of albatrosses to their breeding site between breeding attempts (e.g., Buller’s albatross Thalassarche bulleri; Sagar, Stahl & Molloy, 1998).

Reported recoveries of dead individuals were incorporated into the dataset by fixing the state of these individuals to “dead” in the year they were found.

From the base model, other models were created, by introducing some complexity in survival rates:

• A gender-based difference in survival was tested by estimating an annual survival rate for adult males ϕM relatively to that of females (ϕF): (22) logitϕM=logitϕF+βM.

A uniform prior between 0.7 and 1.0 was used for ϕF, as the mean adult annual survival rate was known to be high (Robertson, 1993), and a non-informative normal prior of mean 0 and standard deviation 100 was used for βM.

• Senescence was introduced in the model of adult survival to assess whether survival decreased with age. This assessment was achieved by modelling the annual survival rate on the logit scale as a linear quadratic function of years survived since the observed minimum age at first reproduction (six years after fledging): (23) logitϕa=logitϕ6for 1≤a<6,logitϕ6+αAa−6+βAa−62for a≥6.

A uniform prior between 0.7 and 1.0 was used for ϕ6, a half-normal prior with a standard deviation of 0.1 was used for αA, and a normal prior of mean 0 and standard deviation of 0.01 was used for βA. The priors for αA and βA were constrained so that the quadratic function on the logit scale did not produce overflow errors (thereby causing the model-fitting to fail).

• Interannual variation in survival rates was introduced in two different ways. First, the annual survival rate was estimated for each year independently, without constraint. In this case, each prior was a uniform distribution between 0.3 and 1.0 for adults, in contrast to previous priors for adult survival (of 0.7–1.0). This approach allowed survival to be unusually low in a given year due to extreme weather for instance. Priors were uninformative (between 0 and 1) for juveniles and pre-breeders. Second, annual survival ϕt was allowed to vary randomly between years, with the variation ϵ around a mean survival ϕ¯ on the logit scale drawn each year t: (24) logitϕt=logitϕ¯+ϵt.

The prior for ϵ was a non-informative normal distribution of mean 0 and precision τ (the inverse of standard deviation), with the hyperparameter τ drawn from a gamma distribution of shape and rate 0.001.

A combination of these modifications was used to create 10 models, including the base model (Model 0) (see Table 1). Models with single effects were not included when the examination of the raw data did not reveal any clear effect (e.g. gender).

Table 1 Models used to estimate annual survival of northern royal albatross at the Taiaroa Head colony.

Each model differed in the way survival was modelled for each life history stages. “Sex” indicates a difference between genders, “age” indicates that survival varied with age (senescence), “year” indicates that survival varied each year independently, “year_re” indicates that annual variations in survival were random, “.” indicates a constant survival rate.

Model	Juvenile	Pre-breeder	Adult	
Model 0	.	.	.	
Model 1	.	.	age	
Model 2	.	.	sex + age	
Model 3	.	.	age + year_re	
Model 4	.	.	sex + age + year_re	
Model 5	year_re	year_re	age + year_re	
Model 6	year_re	year_re	sex + age + year_re	
Model 7	year	year	age + year	
Model 8	year_re	year_re	year_re	
Model 9	.	.	year_re	

The models were coded in the BUGS language (Spiegelhalter et al., 2003), and fitted using the Gibb’s sampling software package JAGS (Plummer, 2005), running two chains, with a burn-in of 50 000 iterations, and a further run of 250 000 iterations. Samples of the model parameters were kept every 100 iterations, in order to prevent auto-correlation within the chains. The chains of the models were checked to verify that the convergence of chains, as well as the existence of potential correlations between and within chains, were satisfactory, by using the “coda” library (Plummer et al., 2006) in the statistical package R (R Development Core Team, 2008).

Models were compared using the Deviance Information Criterion (DIC), which can be approximated from the mean (θ¯) and variance (var(θ)) of the model deviance θ, using the formula (Gelman et al., 2004): (25) DIC=θ¯+12varθ.

The DIC is the Bayesian equivalent of the Akaike Information Criterion (AIC; Akaike, 1974), and represents a trade-off between the deviance and the number of effective parameters (Burnham & Anderson, 2002). A lower DIC value represents a better fit, relative to the degrees of freedom, and like AIC, models receiving a DIC of three units or higher compared to that of the best model may be considered to have considerable less support (Spiegelhalter et al., 2002).

The mean age at first return and mean age at first reproduction were calculated by simulating the fate of a cohort of individuals over time, starting from fledglings in a given year, and calculating the proportion of the surviving individuals of each state year after year by using the mean transition probabilities estimated from the best model. Following Dillingham et al. (2012), the expected age at first return αR and at first breeding αB are respectively: (26) EαR=∑i=1∞iπRi/∑i=1∞πRi,andEαB=∑i=1∞iπBi/∑i=1∞πBi,

with πRi as the unconditional probability that an individual from the cohort first returns to the colony at age i, and πBi the unconditional probability that an individual from the cohort first breeds at age i. The probabilities πRi, and equivalently πBi, were calculated from the simulation as the proportion of the cohort that survived and first returned to the colony (or first bred) at age i, divided by the proportion of the cohort that survived until age i. The uncertainty around E(αR) and E(αB) was calculated by repeating the calculation for each of the 5,000 samples of the parameter’s posterior distributions.

Similarly, for models with senescence, an overall annual adult survival was calculated by taking the average of annual survival rates across all adult ages a, weighted by the probability of reaching age a.

Because all individuals at the Taiaroa Head colony were banded, the total population size could be calculated. This calculation was achieved by summing in the model the number each year of juveniles, pre-breeders, and adults, based on the state of each individual, regardless of their presence at the colony (because the census time is at the start of the breeding period, chicks are not present in the population). From this calculation, the ratios of the total population size to the number of adults and to the number of annual breeding pairs were also calculated.

Data preparation for model fitting and the calculations to derive the parameters of interest were carried out in R 2.15.3 (R Development Core Team, 2008).

A self-contained project containing the data and the code for running the model with random year effect in the survival of the three life history stages, and senescence in adult survival, is available at http://figshare.com/articles/Files_to_run_JAGS_model/1340107.

Results

The most parsimonious model, as judged by the DIC, only included random inter-annual variations in survival rates of all life history stages, and senescence for adults (Model 5) (Table 2). The inclusion of senescence greatly improved the model’s fit. The three models with the lowest deviance and DIC all included an age-dependent adult survival rate; removing it from the most parsimonious model led to an increase in DIC of over 13 between models 5 and 8 (Table 2).

All life history stages showed some random interannual variation in survival rates. Removing this variation for juveniles and pre-breeders (Model 3), as well as for adults (Model 1), led to an increase in the DIC of more than 9. There was no support for a difference in adult survival between genders, as the inclusion of a gender difference in the most parsimonious model led to an increase in DIC of more than 14 (Model 6).

Table 2 Comparison of the models to estimate the annual survival rate of northern royal albatross at the Taiaroa Head colony.

Each model differed in the way survival was modelled for each life history stage, including juveniles SJ, pre-breeders SPB, and adults SA. “Sex” indicates a difference between genders, “age” indicates that survival varied with age (senescence), “year” indicates that survival varied each year independently, “year_re” indicates that annual variations in survival were random, “.” indicates a constant survival rate. Included are the model deviance, the Deviance Information Criterion (DIC), the difference in DIC (ΔDIC) in relation to the most parsimonious model (Model 5).

		Deviance			
#	Model	Mean	Variance	DIC	ΔDIC	
5	SJ(year_re), SPB(year_re), SA(age+year_re)	3079.10	548.40	3353.30	0.00	
3	SJ(.), SPB(.), SA(age+year_re)	3078.90	566.81	3362.31	9.01	
1	SJ(.), SPB(.), SA(age)	3082.64	559.60	3362.44	9.14	
0	SJ(.), SPB(.), SA(.)	3087.27	555.73	3365.13	11.83	
2	SJ(.), SPB(.), SA(sex+age)	3082.64	566.48	3365.88	12.58	
8	SJ(year_re), SPB(year_re), SA(year_re)	3083.95	565.01	3366.46	13.16	
6	SJ(year_re), SPB(year_re), SA(sex+age+year_re)	3078.91	577.59	3367.71	14.41	
9	SJ(.), SPB(.), SA(year_re)	3083.24	569.45	3367.96	14.66	
4	SJ(.), SPB(.), SA(sex+age+year_re)	3078.59	582.92	3370.04	16.74	
7	SJ(year), SPB(year), SA(age+year)	3172.26	747.69	3546.11	192.81	

Under the most parsimonious model, the mean annual juvenile survival was estimated to be 0.933 (95% CI [0.908–0.966]), and that of pre-breeders was estimated at 0.966 (95% CI [0.950–0.980]). The overall annual survival rate for all adults, taking into account the age distribution within this stage, was 0.950 (95% CI [0.941–0.959]). These estimates were robust to the model selection process, as under the second-most parsimonious model, the mean annual juvenile survival was estimated to be 0.927 (95% CI [0.907–0.946]), that of pre-breeders to be 0.965 (95% CI [0.949–0.979]), and that of adults to be 0.950 (95% CI [0.941–0.959]).

The probability of adults breeding the year following a failed attempt was 0.828 (95% CI [0.789–0.864]), in comparison with a probability of 0.006 (95% CI [0.001–0.015]) for adults who bred successfully the previous year . The probability of non-breeding adults breeding the following year was 0.791 (95% CI [0.757–0.823]). Overall, the proportion of adults breeding at the colony in a given year was 56.7% (95% CI [56.5%–56.9%]).

Adults not breeding or breeding unsuccessfully in a given year had a probability of 0.954 (95% CI [0.917–0.979]) of returning to the colony the following year. In contrast, the mean probability of successful breeders returning the following year was 0.162 (95% CI [0.129–0.195]). Almost all pre-breeders returned to the colony year after year; their probability of being present at the colony was estimated to be 0.991 (95% CI [0.982–0.997]).

The mean annual survival rate of adults at six years of age was estimated at 0.976 (95% CI [0.963–0.988]), declining with age to 0.968 (95% CI [0.957–0.979]) at 20 years old, and 0.915 (95% CI [0.879–0.946]) at 40 years old (Fig. 3). The parameters αA and βA defining the shape of senescence were estimated with a mean of −0.032 (95% CI [−0.095–−0.001]) and −0.001 (95% CI [−0.002–0.001]), respectively.

Figure 3 Age-dependent adult survival rate of northern royal albatross at Taiaroa Head, as estimated from the most parsimonious model.

Shown are the mean (line) and 95% credible interval (shading). The points represent the naive estimates from the data, and the error bars the 95% credible interval around the mean obtained from 1,000 bootstraps. The minimum age at first reproduction observed in the population (six years) is represented by the vertical dotted line.

Slight interannual variations in the survival rates occurred in each of the three life history stages, juveniles, pre-breeders and adults, with no detectable trends (Fig. 4). The annual adult survival rate varied between 0.963 (95% CI [0.913–0.985]) in 1990 and 0.980 (95% CI [0.963–0.995]) in 2001. For juveniles, the annual survival rate varied between 0.892 (95% CI [0.592–0.968]) in 2009 and 0.935 (95% CI [0.878–0.988]) in 1998, and for pre-breeders between 0.956 (95% CI [0.881–0.980]) in 1998 and 0.968 (95% CI [0.945–0.990]) in 2008. The inter-annual variance in survival rate (on the logit scale) was estimated at 0.360 (95% CI [0.001–2.432]) for juveniles, 0.129 (95% CI [0.001–0.836]) for pre-breeders, and 0.168 (95% CI [0.001–0.773]) for adults.

Figure 4 Interannual variation of mean annual survival rates for juvenile, pre-breeders and adult northern royal albatross at Taiaroa Head, as estimated by the most parsimonious model, between 1989–90 and 2011–12.

The annual survival for adults refers to individuals that were six years old, the minimum age at first reproduction. Shown are the mean (line) and 95% credible interval around the mean (shading).

The high uncertainty around juvenile survival from 2008 onwards was due to fledglings spending their first years at sea so that they were not subsequently observed returning to the colony. The high uncertainty was also due to a decline in the number of juveniles returning to the colony in 2010–11 and 2011–12 (see Fig. 1). Juvenile survival can, therefore, not be estimated precisely for the last years, as it is confounded by juveniles being away from the colony.

The mean age of first return of juveniles to the colony was estimated at 4.8 years (95% CI [4.6–5.1]), and varied from 3 to 11 years in the Taiaroa Head population (Fig. 5). The mean age at first reproduction was estimated at 8.9 years (95% CI [8.5–9.3]), and ranged from 6 to 16 years after fledging (Fig. 6).

Figure 5 Age distribution of individuals at the age of first return (years since hatching year) for northern royal albatross at the Taiaroa Head colony.

Vertical bars represent the observed distribution from 226 individuals of known age observed at first return to the colony. The line represents the distribution calculated from the most parsimonious model, showing the mean and the 95% credible interval around the mean.

Figure 6 Age distribution of individuals at the age of first reproduction (years since hatching year) for northern royal albatross at the Taiaroa Head colony.

Vertical bars represent the observed distribution from 128 individuals of known age observed breeding for the first time at the colony. The line represents the distribution calculated from the most parsimonious model, showing the mean and 95% credible interval around the mean.

The fit of the model to the data was considered good, as evident in the comparison of the model estimates to the naive age-dependent survival rates (Fig. 3), the distribution of data of the age of first return (Fig. 5), the distribution of data of the age at first reproduction (Fig. 6), and the changes in the relative proportions of life history stages with age (Fig. 7A).

Demographic information provided by the model also included the expected lifespan of the birds. According to the most parsimonious model, half the fledglings may reach an age of 16 years, and 1% of fledglings may reach an age of 53 years (Fig. 7B).

Figure 7 Age distribution of northern royal albatross at Taiaroa Head across different life history stages.

(A) Relative proportion of live birds and (B) probability of individuals being in a given life history stage or dead as a function of age. Presented are mean values and 95% credible intervals obtained from the most parsimonious model (solid line and shading), with (A) including the observed proportions obtained from the data (dashed line).

The population size of northern royal albatross at Taiaroa Head increased linearly over the study period, with the number of individuals present at the colony doubling since 1989–90, and a total population size estimated at 207 (95% CI [194–217]) individuals in 2010–11 (Fig. 8). The total population size was only calculated from 1996–97 onwards, as individuals could only be included in the total following their first capture or resighting in the data. Although most adults banded before 1989–90 were first resighted in 1989–90, five individuals were first resighted in 1990–91 (but none after), and the first resightings of fledglings banded before 1989–90 were spread out from 1989–90 to 1995–96. In addition, the total population size in 2011–12 does not include the fledglings of that year, because the breeding season was not completed at the time of this study.

Figure 8 Population growth of northern royal albatross at Taiaroa Head from 1989 to 2011, including the means and 95% credible intervals for the number of breeding pairs, the number of adults, and the total population size.

Estimation of the total population size was restricted to the period from 1996–97 to 2010–11, as it only included birds since their first resighting or banding after 1989–90 and not all birds alive returned before 1996–97. Also, there was no estimate for 2011–12 as the breeding season was not completed at the time of the study, so that the number of fledglings for this season was still unknown.

During the period from 1996–97 to 2010–11, the ratio of the total population size to the number of breeding pairs varied from 5 to 12 among years (Fig. 9), with a mean value across years of 7.65 (95% CI [7.56–7.78]). The ratio of the total population size to the total number of adults varied from 1.9 to 2.2, with a mean value across years of 2.07 (95% CI [2.04–2.1]).

Figure 9 Annual ratio (mean and 95% credible interval) of the total population size to the number of annual breeding pairs (N/NBP; orange) and to the total number of adults (N/NA; purple) of northern royal albatross at Taiaroa Head from 1996–97 to 2010–11.

Discussion

The self-established colony of northern royal albatross on the mainland of New Zealand at Taiaroa Head provides a unique opportunity for long-term intensive monitoring, and we were consequently able to precisely estimate a number of fundamental parameters driving population dynamics.

In this study, the mean age at first return to the colony was estimated to be between four and five years (4.8; 95% CI [4.6–5.1]), close to the previous estimate of four years (Robertson, 1993). It is, however, slightly lower than that of similar species; for example, the age at first return was found to range from five to seven years for Antipodean and Gibson’s albatrosses (Walker & Elliott, 2002), and five to seven years for wandering albatross Diomedea exulans (Pickering, 1989).

The mean age at first reproduction of northern royal albatross was estimated here at 8.9 years (95% CI [8.5–9.3]), which is within the range of the previous estimate of 8 to 10 years (Robertson, 1993). It is, however, lower than the age at first reproduction of similar species, which ranges from 10 to 13 years for Antipodean and Gibson’s albatrosses (Walker & Elliott, 2002), and from 9 to 10 years for wandering albatrosses (after the initial decrease during the study due to density dependence; Weimerskirch, Brothers & Jouventin, 1997).

Using resighting data of northern royal albatross collected from 1937 to 1993, Robertson (1993) estimated the annual adult survival rate to be 0.946. This earlier estimate is within the range of estimates derived from the present study of 0.950 (95% CI [0.941–0.959]), based on data from 1989–2012. Robertson (1993) used simple ratios of the numbers of birds known to be alive over consecutive years, not taking into account that birds may be alive but not present at the colony. As a consequence, the survival ratios he calculated would underestimate the actual survivorship of birds. Here, by using a multi-state capture-recapture model, the probability of birds being at the colony was estimated separately from the survival rates, resulting in a more reliable estimate. To investigate any changes in survival over the whole period since banding began in 1938, it is recommended that all data be analysed within the same framework.

Annual survival rates estimated here are of local survival, i.e., the survival of individuals staying at the colony, with permanent emigration being treated as mortality. As the only other colonies are on the Chatham Islands, about 1,000 km from Taiaroa Head, this distance is likely to impede emigration. Adults are highly philopatric once they choose a breeding site, and the only cases of breeding dispersal are over small distances, as observed for black-browed (Thalassarche melanophris) and grey-headed (T. chrysostoma) albatrosses (Prince et al., 1994), southern Buller’s albatross (T. bulleri bulleri; Sagar, Stahl & Molloy, 1998), and wandering albatross (Inchausti & Weimerskirch, 2002). Juvenile dispersal is more likely than adult emigration, but the probability of dispersing over 1,000 km is low (Inchausti & Weimerskirch, 2002). Only 16 individuals recorded at Taiaroa Head have immigrated from the Chatham Islands since 1989. This immigration reflects a low rate of movement, considering the size of the Chatham Islands population (over 5,000 annual breeding pairs). Juvenile dispersal from Taiaroa Head has been recorded; one banded chick from Taiaroa Head has been subsequently recorded as breeding on the Chatham Islands. The monitoring effort of northern royal albatross at the Chatham Islands has been low and sporadic, and it is possible that more birds from the Taiaroa Head colony have emigrated there without being detected. Because of the potential for migration, juvenile survival may be underestimated.

The colony at Taiaroa Head is extensively managed, and it is possible that the survival rates of northern royal albatross breeding at Taiaroa Head differ from those at the Chatham Islands. Birds at Taiaroa Head benefit from active management that aims to maximise the survival of adults and chicks. In addition, birds from the two colonies may forage in different areas, exposing them to different threats. Environmental conditions in early stages of life may also impact individuals’ fitness (Lindström, 1999). Nevertheless, a simple estimate of adult survival of northern royal albatross on Chatham Islands was 95.2% (ACAP, 2009), within the range of estimates for adult survival at Taiaroa Head.

The population of northern royal albatross at Taiaroa Head has been steadily increasing since its inception, and doubled from 1989–90 to 2011–12, with a total population size estimated to be 207 individuals (95% CI [194–217]) in 2010–11. This total does not include four immigrants from the Chatham Islands that were seen alive at Taiaroa Head in 2010–11 and 2011–12 but not included in the analysis because of their unknown age. The increase in the population has been linear, and the ratio of the total population size to the number of breeding pairs has not shown any trend over time (Fig. 9), suggesting that the population dynamics do not appear to be subject to density dependence.

Although the population size at Taiaroa Head has been increasing (see Figs. 1 and 8), some synchronicity in attendance occurred from 2002–03 onwards, as a result of higher-than-normal attendance combined with high breeding success, as well as a low breeding success the previous year. Biennial breeding led to large inter-annual variations in colony size, with 42% fewer birds present at the colony in 2003–04 compared with 2002–03. Although the breeding success of the Taiaroa Head colony is managed, which may have contributed to the high breeding success in 2002–03, the succession of a “bad” year followed by a “good” year may be common in less actively managed seabird populations, as seen in the high inter-annual variability in survival rates found in some long-term studies (e.g., Jenouvrier, Barbraud & Weimerskirch, 2003; Frederiksen et al., 2008). Population estimates for seabirds rely on census data, most often consisting of counts of the number of breeding pairs present at colonies, from which population trends over time are often derived (e.g., IUCN, 2013). Because of logistical and financial constraints, surveys may be carried out only in a single year (e.g., Sutherland & Dann, 2012), or not in successive years (e.g., Weimerskirch, Zotier & Jouventin, 1989; Ryan et al., 2003; Francis & Sagar, 2012). In these cases, large inter-annual population fluctuations due to changes in bird attendance may lead to large biases in population estimates.

While the number of annual breeding pairs may sometimes be sufficient for detecting population trends (IUCN, 2013), the estimation of the total population size may be sometimes required, e.g., for estimating the demographic impact of threats (Dillingham & Fletcher, 2011; Waugh et al., 2012; Richard & Abraham, 2013b) or the food consumption of seabird populations (de L Brooke, 2004b). In these cases, multipliers for estimating the total population size from the number of annual breeding pairs are necessary. Because all individuals at Taiaroa Head are banded, we were able to estimate the number of individuals of each life history stage, while also estimating the ratio of the total population size to the number of breeding pairs at 7.65 (95% CI [7.56–7.78]), and the ratio of the total population size to the number of adults at 2.07 (95% CI [2.04–2.1]). To our knowledge, these ratios are the only direct estimates available for biennially breeding albatrosses. The ratios used in the literature are from expert opinion (Gales, 1998; de L Brooke, 2004a; Waugh et al., 2012), simulations (Gilbert, 2009; Richard & Abraham, 2013a), or allometric relationships (Dillingham & Fletcher, 2011). For the ratio of the total population size to the number of annual breeding pairs, published values are 5 (de L Brooke, 2004b; Waugh et al., 2012), 6.5 (de L Brooke, 2004a), 8.3 (Richard & Abraham, 2013a), and 10.3 (Dillingham & Fletcher, 2011). Due to large inter-annual fluctuations in the number of breeding pairs at the colony, this ratio was highly variable among years, varying from 5 to 12. However, because the age distribution in a population is related to population growth (Caswell, 2001), the representativeness of this ratio for other populations is unclear, due to the ongoing management at Taiaroa Head.

Northern royal albatross are long-lived, with one bird at Taiaroa Head known to have survived at least 51.5 years since banding (and likely 61 years since fledging; Robertson, 1993). Our results support the decline in survival rate with age that Robertson (1993) detected for birds over 25 years of age using naive survival estimates. Senescence has been found in other albatross species, with survival declining from similar ages, after 28 years in wandering albatross (Weimerskirch, 1992), and after 25 years in southern Buller’s albatross (Sagar et al., 2000).

A number of assumptions were made in the models in order to keep the models simple to be able to be fitted. For instance, the transition between breeding and non-breeding for adults was assumed to be constant over time and with respect to age or gender. Similarly, the transition between life history stages were not gender dependent. More data in the future may allow more complex models to be fitted, and to test hypotheses about factors affecting each demographic processes in finer details.

Long-term monitoring of long-lived albatrosses is crucial for robust estimation of the essential aspects of their life history. In particular, the estimation of age at first return and age at first reproduction may be negatively biased when monitoring is not carried out for a sufficient length of time (Dillingham et al., 2012). Intensive long-term monitoring of albatross colonies is rare due to their remoteness (but see e.g., Wooller, Bradley & Croxall, 1992; Jenouvrier, Barbraud & Weimerskirch, 2003). The mainland colony of northern royal albatross at Taiaroa Head has been monitored since its inception, providing a unique opportunity for understanding albatross demographics. Capture-recapture data from 21 years has shown that survival rates of the Gibson’s albatross Diomedea antipodensis gibsoni, also breeding in New Zealand, have declined (Francis, Elliott & Walker, 2011). From the current analysis, there is no indication of a similar decline in the survival rate of northern royal albatross from 1998–99 to 2011–12. Ongoing monitoring of the Taiaroa Head population will allow for any similar changes in the survival rate of northern royal albatross to be detected.

We are very grateful to Peter Dillingham for providing us with the WinBUGS code of his multi-state model of capture-recapture of Gibson’s albatrosses, and Sebastián Jiménez for his reports of albatross captures in Uruguayan longline fisheries. We are also grateful to Johanna Pierre and Katrin Berkenbusch for their comments and edits of the manuscript, as well as to the two reviewers. The Taiaroa Head colony would probably not have persisted and flourished without the help of a number of people, particularly the late L.E. Richdale, S. Sharpe, A. Wright, and other former and current staff. Finally, the technical completion of this work has been dependent on open-source software and packages, especially R, JAGS, Latex, Emacs, and Ubuntu. We are extremely grateful to the many people who contribute to these software projects.

Additional Information and Declarations

Competing Interests

Author Contributions

Data Deposition

Yvan Richard and Edward R. Abraham are employees of Dragonfly Science and Lyndon Perriman is an employee of the New Zealand Department of Conservation.

Yvan Richard analyzed the data, contributed reagents/materials/analysis tools, wrote the paper, prepared figures and/or tables, reviewed drafts of the paper.

Lyndon Perriman and Chris Lalas reviewed drafts of the paper, expertise; Data.

Edward R. Abraham contributed reagents/materials/analysis tools, reviewed drafts of the paper.

The following information was supplied regarding the deposition of related data:

http://figshare.com/articles/Files_to_run_JAGS_model/1340107.

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
