# Peer review of "Demographic rates of northern royal albatross at Taiaroa Head, New Zealand"

_PeerJ, doi:10.7717/peerj.906_

## Round 0.1 · original submission · Major Revisions

The paper is interesting and generally well written. The Methods are quite confusing, and some clarifications have to be made. The authors should follow the recommendations made by both reviewers.

Also some results need more explanation.

·

Basic reporting

The manuscript is generally well written and presented.

Experimental design

The statistical methods used in this manuscript are generally appropriate and of a high standard. My comments in this area mainly regard the description of the methods.
- The authors stress that the statistical model used is Bayesian, presenting this as a major thing. However, the really important point in my opinion is that it is a multi-state mark-recapture model. Such a model could also have been fitted in a non-Bayesian context (e.g. in E-SURGE).
- The model is described as having three age classes, each covering several years of age. This terminology is potentially quite confusing, and I would suggest renaming these classes 'life history stages' or something similar.
- It seems to be an underlying assumption of the model that the probability of observing an individual, given that it is present in the colony, is 1. This is very rarely true in field studies, but I'm prepared to accept that it is the case here. However, the assumption should be stated clearly.
- Not all the priors of the basic model are listed.
- Estimates from Bayesian models do not have confidence limits, but credible limits. Please change the wording throughout.

Validity of the findings

I see no indication that the data underlying the manuscript have been or will be provided, as required. In addition, I would strongly suggest that the authors make the WinBUGS code available.

Additional comments

I have only few and minor comments:
- Abstract, first sentence: this statement is not quite true, seabird studies have supplied many of the best series of estimates of demographic rates.
- Results: nearly all estimates presented in Table 3 are also listed in the text, so the table seems unnecessary. The exception is the estimates of process variance of survival of the three stages, which perhaps should be mentioned in the text as well as such estimates are rare and quite interesting (highest for juveniles, as expected). By the way, there are some very small differences between the estimates presented in the text and in Table 3, probably rounding errors.
- Near the bottom of p. 8, there are two references to Table 3 - one should be to Figure 3.
- At the beginning of the discussion, the estimates of mean age of return and first breeding are repeated, but with more decimals than in the results section. This is a bit weird.
- Bottom of p. 14: I don't think Frederiksen et al (2008) mentions that bad years are succeeded by good years, rather that variance in survival can be very high for some species. You might refer to this paper in a discussion of process variance.
- References: the authors 'de l. Brooke' and 'Brooke' are one and the same. Thus, there should be a Brooke 2004a, and a Brooke 2004b

Reviewer 2 ·

Basic reporting

The keyword “Taiaroa Head” doesn’t seem very useful, or maybe very important. It might be more useful to use something like “demographics” or “age structure” or “population size/trend/growth” instead.

I’m not familiar with DIC. Is there a rule of thumb for comparing values of delta-DIC, equivalent to delta-AIC (delta-AIC < 2 indicates no better support for one model over another)?

I would like to see the number of parameters for each model added to Table 2.

Table 3: Include P(breed|pre-breeder) and P(pre-breeder|juvenile)?

Does total population size include nestlings or just chicks that fledge?

Typos:
The last sentence before equation 25 should have “of” inserted between “the formula” and “Gelman et al.”.
The last sentence of that paragraph should say “A lower DIC value”, not “A lower DIC values”.
The 6th paragraph of Results ends with “(Table 3, Table 3)”. I think it should be “(Table 3, Fig. 3)”.
The 4th complete paragraph on P. 14 gives 95% c.i. as 180-205. In Results, they are 194-217, which makes a lot more sense because the estimate is 207.
The 4th line on P. 15: drop the “s” from “seabirds”.

Experimental design

What does running two chains mean? Convergence of chains and correlations between and within chains were assessed, but there’s nothing in results about them. What’s the significance of any convergence or correlations? Were there any?

What’s the difference between using every 100th sample from 250,000 iterations and just using all samples from 2500 iterations? Did you verify that your program can generate 250,000 pseudo-random numbers without repeating sequences?

I would like to see some justification for the 10 models tested. Why not a model with just sex or just year or sex + year?

It looks like only survival was allowed to vary. How good an assumption is it that transition and resighting probabilities don’t vary with time or age or sex?

Validity of the findings

Fig. 1 and Fig. 8 don’t seem to match. Fig. 1 shows number of breeding adults, mostly between about 30 and 60, and Fig. 8 shows number of breeding pairs, which should be half the number of breeding adults. However, it looks like the numbers in Fig. 8, all < 20-25, are lower than half the numbers in Fig. 1.

The 1st sentence of Results says that the most parsimonious model was a simple one. But I think it has more parameters than most of the other models. Why is it simple?

Fig. 8 and Fig. 9 don’t seem to match. Specifically, N/NBP in Fig. 9 seems lower than it should be based on the numbers for total population and breeding pairs in Fig. 8.

The 4th complete paragraph on P. 14 states that there does not appear to be density dependence. But it appears from Fig. 8 that the number of breeding pairs has not increased as much as the population or number of adults. Is that just because the scale is compressed for the lower numbers? If mates, nest sites, or some resource essential for breeding are limiting, the total population could increase without an increase in the number of breeding pairs, and it may indicate density dependence.

Additional comments

It would be easier to review the manuscript if it had line numbers.

---

## Round 0.2 · accepted · Accept

Authors incorporated the main suggestions made by the reviewers. The description of the Methods was improved, confusing wording was changed (pe: age class to life story age), the Table was removed, and discussion improved. Also minor editing changes were done.